# A Compensation Method for Full-Field-of-View Energy Nonuniformity in Dark-and-Weak-Target Simulators

**DOI:** 10.3390/s24134147

**Published:** 2024-06-26

**Authors:** Fenghuan Quan, Shi Liu, Gaofei Sun, Jian Zhang, Yu Zhang, Bin Zhao, Jierui Zhang

**Affiliations:** Aerospace Ground Simulation Test and Testing Technology Institute, Changchun University of Science and Technology, Changchun 130012, China; quanfenghuan@custa9.wecom.work (F.Q.); 2021100325@mails.cust.edu.cn (G.S.); zhangjian@cust.edu.cn (J.Z.); yuzhang@mails.cust.edu.cn (Y.Z.); zb@mails.cust.edu.cn (B.Z.); 2020100317@mails.cust.edu.cn (J.Z.)

**Keywords:** dark-and-weak-target simulation, adaptive compensation of response error, improvement of uniformity

## Abstract

Dark-and-weak-target simulators are used as ground-based calibration devices to test and calibrate the performance metrics of star sensors. However, these simulators are affected by full-field-of-view energy nonuniformity. This problem impacts the quality of output images and the calibration accuracy of sensors and inhibits further improvements in navigational accuracy. In the study reported in this paper, we sought to analyze the factors which affect full-field-of-view energy uniformity in dark-and-weak-target simulators. These include uneven irradiation in backlight sources, the leakage of light from LCD display panels, and the vignetting of collimating optical systems. We then established an energy transfer model of a dark-and-weak-target simulator based on the propagation of a point light source and proposed a self-adaptive compensation algorithm based on pixel-by-pixel fitting. This algorithm used a sensor to capture the output image of a dark-and-weak-target simulator and iteratively calculated the response error matrix of the simulator. Finally, we validated the feasibility and effectiveness of the compensation algorithm by acquiring images using a self-built test system. The results showed that, after compensating an output image of the dark-and-weak-target simulator, the grayscale standard display function (SDF) of the acquired sensor image was reduced by about 50% overall, so the acquisition image was more accurately compensated, and the desired level of grayscale distribution was obtained. This study provides a reference for improving the quality of output images from dark-and-weak-target simulators, so that the working environments of star sensors may be more realistically simulated, and their detection performance improved.

## 1. Introduction

With the continuing development of deep-space exploration technology, the number of launches of various types of spacecrafts is steadily increasing. At the same time, there is a demand for ever more accurate information from spacecraft. In spacecraft, the star sensor is the main attitude measurement device [1,2]. The measurement accuracy of the star sensor is crucial to the navigational accuracy of the spacecraft. In order to improve the measurement accuracy of a star sensor, a simulator must be used so that the sensor may be calibrated on the ground. However, the minimum magnitudes of background stars generated by existing simulators are much lower than the limiting detection magnitudes of existing star sensors, and the irradiation of the image plane is nonuniform. This means that the maximum detection capability of star sensors cannot be calibrated, and this inhibits any improvements in navigational accuracy. There is, therefore, an urgent need for a dark-and-weak-target simulator that can simulate the illumination of faint stars while maintaining energy uniformity in the image plane, so that the uniformity of radiated light displayed in star charts is maintained, and simulation input conditions are provided which meet the requirements for ground calibration of the star sensor.

To enhance full-field-of-view energy uniformity in dark-and-weak-target simulators, researchers have sought to optimally adjust the structures of light sources and improve the performance of optical systems. In some published studies [3,4], researchers have reported the use of free-form lenses, plane mirror arrays, and interferometric structures to optimally adjust the structures of light sources, for the purpose of improving full-field-of-view energy uniformity. However, when light from a light source hits an optical system, there is a greater impact on the energy uniformity of the image surface as a result of the vignetting effect. In other studies, therefore [5,6], researchers have optimized and improved optical systems using Lambert’s cosine theory and aberration theory, resulting in improved output-image quality. Still, other researchers [7] have described improvements in the transmission of light through new designs of hair-glass material in optical systems.

Although the uniformity of illumination in the image plane of a dark-and-weak-target simulator can be improved by optimizing the design of the light source and the optical system, image quality and information extraction may be further improved by the use of image-processing algorithms. Existing image-processing algorithms [8] are based on different techniques such as response curve-fitting [9,10] and frequency-domain analysis [11]. By such means, the physical properties and response functions of the system which result in vignetting may be modeled, and a correction method adopted, such as coordinate transformation [12], mathematical fitting [13], marginal distribution estimation [14], or parabolic fitting [15]. All these methods are designed to compensate for the system’s vignetting effect and improve image uniformity. However, such methods are either computationally complex or result in an inadequate fit, and most rely on additional information to achieve effective performance.

In the present study, we sought to address the problem of full-field-of-view energy uniformity affecting dark-and-weak-target simulators. By modeling energy transfer in such a simulator, we developed an adaptive compensation algorithm based on pixel-by-pixel fitting. We also established an experimental platform to experimentally validate the compensation method and found that it achieved effective compensation of full-field-of-view energy nonuniformity in dark-and-weak-target simulators.

## 2. Composition and Working Principle of the Dark-and-Weak-Target Simulator

The composition of the dark-and-weak-target simulator is shown in Figure 1. It can be seen that it comprised an attitude-and-orbit-dynamics simulation control computer, an LCD display driver circuit, a backlight light source, an LCD display, and a collimating optical system.

The attitude-and-orbit-dynamics simulation control computer, in conjunction with the target parameter settings and the mission requirements, was used to calculate and output image content in real time, including position information for the dark/weak targets, as well as environmental information. During output to the LCD display panel, by controlling the output image brightness L of the LCD display panel, and the output brightness B of the LEDs in the backlight light source, we increased or decreased the overall brightness of the final output image to harmonize and optimize the final imaging effect, while satisfying the following relationships:(1)$$\left\{\begin{array}{l} B=B_{0}\beta \\ L=L_{0}\alpha \end{array}\right.$$
where $B_{0}$ and $L_{0}$ are the initial brightness of the backlight source and LCD display panel, respectively, and β and α are adjustable parameters. Synergistic control between the two units was achieved through image-sensing and feedback mechanisms. When the output image brightness was weak, we increased the values of β and α; when the output image brightness was excessive, we lowered the values of β and α. This adjustment completed the multi-stage optimization of the output image brightness.

The image content output from the attitude-and-orbit-dynamics simulation control computer was collimated by an optical system, to project positional and environmental information from dark/weak targets onto a specified light-sensitive surface. The output image content under different parameters could be optically mapped to achieve a high degree of simulation, and effective visualization of a variety of dark/weak targets, to fulfil the requirement for a wide range of dark/weak targets, and thereby display the effect of simulation.

## 3. Energy Transfer Modeling and Analysis of Influencing Factors

### 3.1. Energy Transfer Model

The process of energy transfer from a pixel point on an LCD display panel to an image plane can be understood as a mapping relationship in which a pixel point n∆p on the LCD display is eventually mapped to a corresponding position m∆p on the image plane. On this basis, each pixel point on the LCD display panel can be regarded as a point light source, and a transfer function can be used to represent changes in the light ray during the propagation process. We therefore established an energy transfer model, as shown in Figure 2. We then calculated the light intensity received at different positions on the image plane using Equation (2), and finally obtained the light intensity on the image plane I(m∆p).
(2)$$I(m∆p)=\frac{1}{f^{2}}\int_{m∆p-∆p/2}^{m∆p+∆p/2}E(\lambda )\alpha (n∆p)L(n∆p)(1+∆L(n∆p))T(r-n∆p,\lambda ,f))dr$$

In Equation (2), ∆p is the width size of the pixel, E(λ) is the luminous intensity of the backlight source, α(n∆p) is the modulation factor of the LCD display panel for the nth pixel, L(n∆p) is the transmittance function of the nth pixel of the LCD display panel, ∆L(n∆p) is the amount of variation in light leakage between the pixels, f is the focal length of the collimating optical system, r represents the radial coordinates of the transmission of the light rays, and T(r−n∆p,λ,f) is the transfer function of the light rays through the optical system.

With the energy transfer model established using the above equation, an integral value for light intensity at each position could be calculated. By repeating this process for all positions, the distribution of light intensity across the whole image plane could be obtained.

### 3.2. Analysis of Influencing Factors

From the above description of the energy transfer model, it can be understood that the light intensity received at each position on the image plane is affected by a variety of factors, such as the luminous intensity of the backlight source, the intensity of the light transmitted by each pixel in the LCD display panel, and the efficiency of the optical system in transmitting light. However, due to uneven irradiation in the backlight source, leakage of light from the LCD display panel [16], and vignetting of the collimating optical system, the image-plane energy of the dark-and-weak-target simulator is not uniform. In addition, the transfer process can also be affected by variations in temperature. However, the operating temperature of the dark-and-weak-target simulator can be controlled by physical cooling, to ensure that it operates properly at operating and storage temperatures.

In summary, when the light source and collimating optical system are determined, the luminous intensity of the light source and the transfer function of the light after passing through the optical system will also be determined, and uniquely, so the image-plane energy of the dark-and-weak-target simulator can be adjusted and controlled using the LCD display panel.

## 4. Adaptive Compensation Algorithm Based on Pixel-by-Pixel Fitting

For the present study, we introduced an adaptive compensation algorithm based on pixel-by-pixel fitting. Provided that the uniformity of the light source satisfied our study requirements, the algorithm was able to use sensors to acquire output images with different gray levels; these were displayed by the dark-and-weak-target simulator. The response error surface of the dark-and-weak-target simulator could then be calculated, based on the acquired images. The brightness of each pixel of the dark-and-weak-target simulator was adaptively compensated and adjusted using the error surface, and the compensated output image was then acquired. Finally, grayscale standard display function (SDF) values [17] in the captured images before and after compensation were calculated, so that any improvement in uniformity in the light intensity distributed on the image surface of the dark-and-weak-target simulator could be evaluated. The overall compensation process is set out in flowchart form in Figure 3.

Initial working conditions were established, the uniformity of the light source was measured, and images were captured. It is worth noting that the specific region in which a sensor can ultimately detect a dark-and-weak-target simulator depends on the field of view $\omega_{1}$ of the simulator and the field of view $\omega_{2}$ of the sensor, and these are usually not the same. In order to ensure the complete detection of the dark-and-weak-target simulator by the sensor, the field-of-view angle of the sensor should be greater than or equal to that of the simulator, i.e., $\omega_{2}\geq \omega_{1}$. This may be demonstrated using Equation (3), as follows:(3)$$\left\{\begin{array}{l} \omega_{1}=2arc\tan(\frac{h_{1}}{f_{1}})=2arc\tan(\frac{\sqrt{{\left(np_{1}\right)}^{2}+{\left(mp_{1}\right)}^{2}}}{f_{1}})\\ \omega_{2}=2arc\tan(\frac{h_{2}}{f_{2}})=2arc\tan(\frac{\sqrt{{\left(np_{2}\right)}^{2}+{\left(mp_{2}\right)}^{2}}}{f_{2}}) \end{array}\right.$$where $f_{1}$ is the focal length of the dark-and-weak-target simulator, $f_{2}$ is the focal length of the sensor, $h_{1}$ is the image height of the dark-and-weak-target simulator, $h_{2}$ is the image height of the sensor, $p_{1}$ is the image-element size of the dark-and-weak-target simulator, $p_{2}$ is the image-element size of the sensor, n is the number of transverse elements, and m is the number of longitudinal elements. Thus, by calculating the field-of-view angles of the simulator and the sensor, the percentage of the simulator’s field of view that can be detected by the sensor may be identified. In other words, by detecting localized areas within the imaging range of the dark-and-weak-target simulator and using these as a microcosm of overall system performance, the distribution and spread of dark-and-weak-target simulator imaging over the detection area of the sensor can be accurately described.

The uniformity of the backlight source can be evaluated by measuring the irradiance inhomogeneity of the backlight source using Equation (4), as follows:(4)$$\varepsilon =\frac{∆E_{i}}{\bar{E_{i}}}=\pm \frac{E_{i\max}-E_{i\min}}{E_{i\max}+E_{i\min}}\times 100\%$$
where $E_{i\max}$ is the maximum value of irradiance within the irradiated surface, $E_{i\min}$ is the minimum value of irradiance within the irradiated surface, and $\bar{E}$ is the average irradiance. When irradiance inhomogeneity in the backlight source is better than ±10%, as specified in the CIE standard [18], it can be assumed that the impact of the inhomogeneity of the backlight source on the system is controlled within an acceptable range.

2.Error fitting was performed while ensuring that the uniformity of the backlight source met the study requirements. We let the dark-and-weak-target simulator output 256 images in a grayscale range from 0 to 255, and allowed the initial grayscale matrix of the ith output image to be $G_{i}(x,y)$, and the grayscale of each pixel to be $g_{xy}$, with x and y being the coordinates of the horizontal and vertical positions, respectively, for each pixel point), so that


(5)
$$G_{i}=\left(\begin{array}{cccc} g_{11} & \ldots & g_{1y} & g_{1n}\\ \vdots & \ddots & \vdots & \vdots \\ g_{x1} & & g_{xy} & g_{xn}\\ g_{m1} & \cdots & g_{iy} & g_{mn} \end{array}\right)$$


The grayscale matrix of the ith captured image of the sensor was then denoted as $S_{i}(x,y)$, and the grayscale value of each pixel denoted as $s_{xy}$, so that
(6)$$S_{i}=\left(\begin{array}{cccc} s_{11} & \ldots & s_{1y} & s_{1n}\\ \vdots & \ddots & \vdots & \vdots \\ s_{x1} & & s_{xy} & s_{xn}\\ s_{m1} & \cdots & s_{iy} & s_{mn} \end{array}\right)$$

Next, we calculated the grayscale mean of matrix $S_{i}(x,y)$ to obtain the grayscale mean matrix of the ith captured image. We noted this as $A_{i}(x,y)$, and the grayscale of each pixel as $a_{xy}$, so that
(7)$$A_{i}=\left(\begin{array}{cccc} a_{i} & \ldots & a_{i} & a_{i}\\ \vdots & \ddots & \vdots & \vdots \\ a_{i} & & a_{i} & a_{i}\\ a_{i} & \cdots & a_{i} & a_{i} \end{array}\right)$$
(8)$$a_{i}=\frac{1}{mn}\times \sum_{x=1}^{m}\sum_{y=1}^{n}S_{i}(x,y)$$

We then calculated and superimposed the mean value of the difference between the grayscale matrix of the captured image and the grayscale mean matrix. This value was denoted as $\mu_{i}$, as shown in Equation (9):(9)$$\mu_{i}=\frac{\sum_{i=1}^{n}\left(S_{i}-A_{i}\right)}{i}(i=1,2,\cdots ,n)$$

It should be noted that, when $S_{i}-A_{i}<0$, then the actual response of the pixel is lower than the ideal average response. In such an event, the grayscale of the pixel should be brightened, to make it closer to the ideal average grayscale distribution. If ${∆}_{i}$ is then taken to be the relative–difference matrix between the ith superposition and the i−1th superposition, and ${∆}_{i\max}$ be the maximum relative–difference matrix, then
(10)$${∆}_{i}=\mu_{i}-\mu_{i-1}$$
(11)$${∆}_{i\max}=\max(\;\left|{∆}_{i}\right|\;)$$

A contrast-sensitivity function curve for the human eye reveals that the luminance threshold for visual sensitivity in the human eye is about 1.2% [19]. However, the luminance-detection sensitivity of the sensor used in the present study was much higher than that of the human eye. Consequently, when ${∆}_{i\max}$ satisfied the conditions of Equation (11), it could be assumed that the sensor was unable to distinguish any difference in luminance, and iteration could be terminated.
(12)$$\left\{\begin{array}{l} {∆}_{i\max}\leq 0.12\%\\ {∆}_{(i+2)\max}\leq {∆}_{(i+1)\max}\leq {∆}_{i\max} \end{array}\right.$$

Finally, we denoted the difference–mean matrix $\;\mu_{i+2}$ obtained from the i+2th cycle as the response error matrix of the dark-and-weak-target simulator, denoted as P(x,y).

To determine the uniformity compensation for pixel brightness in the dark-and-weak-target simulator, we obtained the compensated image $G_{i}^{\prime }(x,y)$ by taking the output-image matrix $G_{i}(x,y)$ of the dark-and-weak-target simulator and subtracting the product of the error matrix P(x,y) and the output-image matrix $G_{i}(x,y)$ of the simulator, as shown in Equation (13), as follows:(13)$$G_{i}^{\prime }(x,y)=G_{i}(x,y)-G_{i}(x,y)*P(x,y)$$

Having acquired a second, compensated image, we were able to bring both the initial and compensated images into Equation (14). The grayscale SDF of the acquired image before and after compensation was then calculated, in order to analyze any improvement in the uniformity of light-intensity distribution on the image surface of the dark-and-weak-target simulator, before and after compensation, so that
(14)$$std=\sqrt{\frac{1}{MN}\sum_{x=1}^{M}\sum_{y=1}^{N}\left(S_{i}^{\prime }(x,y)-\bar{S_{i}}\right)}$$
where M and N are the numbers of rows and columns, respectively, in the image, $S_{i}^{\prime }(x,y)$ is the grayscale value of the captured image before and after compensation at the coordinates of point (x,y), and $\bar{S_{i}}$ is the average grayscale value of the image. The lower the grayscale SDF value, the more concentrated and uniform the grayscale distribution in the image. However, this also means that the grayscale distribution in the image is more scattered and uneven. Because the algorithm was designed to fit the combined effect of various influencing factors into a unified response error surface, this became the ‘total factor’ affecting the uniformity of the picture, and the surface was utilized to inversely compensate for the dark-and-weak-target simulator. Any change in distribution before and after compensation could therefore be evaluated using SDF, which actually, if indirectly, reflects the combined effect of the various influencing factors being eliminated.

It can be seen that, through the use of image pixel distribution to reflect the characteristics of light intensity distribution, the use of compensation algorithms to obtain the response error surface, and the use of the dark-and-weak-target simulator image for inverse compensation, an image with uniform grayscale distribution may be obtained for the boundary region. Any distortion and nonuniformity is thereby alleviated, so that the boundary region of the grayscale distribution is more smooth and centralized. The algorithm does not need to accurately model the boundary region of the various factors affecting the complex model, it only needs to obtain a sufficient number of sampling data that can be simplified in the processing at the same time. However, through the appropriate expansion of the sampling region, to enhance the field of view of the boundary region, the energy compensation of the dark-and-weak-target simulator image surface is ultimately achieved.

## 5. Experimental Validation and Results

To verify the feasibility of the compensation method, and as shown in Figure 4, a test environment was constructed by placing the LCD with a backlight source inside the head of the dark-and-weak-target simulator. Control of the LCD panel displaying the image was completed by driving the display circuit. Specifically, an Epson L3P14U-51G00R LCD display panel was used. For the sensor, an FLIR Black S BFS-PGE-50S5M-C camera was used, with an exposure time of $1.36119\times e^{6}\mu s$ and an exposure gain of 2 db.

### 5.1. Establishment of Initial Conditions

The initial conditions were set as follows: since the sensor needs to be calibrated on the ground under dark conditions without stray light interference, a dark room environment was built to isolate the ambient stray light except for the experimental instruments and equipment. At the same time, simulator image element size $p_{1}$ = 0.014 mm; optical system focal length $f_{1}$ = 238.19 mm; field of view $\omega_{1}$ = 7.4°; sensor image element size $p_{2}$ = 0.0074 mm; optical system focal length $f_{2}$ = 111.12 mm; and field of view $\omega_{2}$ = 24°. These conditions were set to satisfy the requirement that $\omega_{2}\geq \omega_{1}$, so as to ensure that the sensor on the dark-and-weak-target simulator effectively achieved an imaging area of complete detection. Irradiation inhomogeneity in the backlight source was measured to determine whether it met the uniformity requirements of the dark-and-weak-target simulator with respect to the backlight source.

With the backlight light source activated, we placed the irradiance meter on the irradiance surface of the light source. We adjusted the current of the backlight power supply so that it reached a maximum value when the light-sensitive surface was facing the backlight light source. When the light source was stabilized, the irradiance meter was used to take rapid measurements at 16 different positions on the surface of the light source, to eliminate any errors arising from instability in the light intensity. As a side-entry backlight panel was used, and default LED beads were placed above the backlight panel, the horizontal irradiation did not change much, while vertical irradiance decreased incrementally due to light attenuation on the propagation path. Figure 5 shows a schematic diagram of the method used for testing irradiance inhomogeneity. Figure 6 shows the irradiance meter used.

The results of the irradiation inhomogeneity testing are shown in Table 1. Irradiation inhomogeneity was calculated by bringing the test results into Equation (4); these data were normalized and plotted to show variations in illuminance corresponding to different positions on the irradiated surface, as shown in Figure 7.

It can be seen that the relative illuminance of the backlight source decreases incrementally. This result was mainly due to the fact that the LED was placed on the side of the backlight source, and light attenuation occurred during the transmission process. The irradiance inhomogeneity of the backlight source was found to be 3.62%, and thus satisfied the requirements of the CIE standard for irradiance inhomogeneity, being better than ±10%.

### 5.2. Error Fitting

Dark-field correction was used to compensate for the dark current of the sensor, and normal operation of the LCD at operating and storage temperatures was ensured by hardware cooling under constant temperature conditions, so that the dark-and-weak-target simulator output a total of 256 images $G_{i}(x,y)$ recorded in 0–255 grayscale, and the sensor was used for one-by-one acquisition to obtain $S_{i}(x,y)$. The initial acquisition of the image of the grayscale SDF curve is shown in Figure 8. In the figure, it can be seen that with the enhancement of the gray value of the image, the larger the grayscale SDF value of the image, and the more uneven the image-plane energy. Initial-acquisition images with grayscale values of 0, 109, and 255 are shown in Figure 9, in which a gradual weakening trend from the center to the edges is clearly evident, along with irregularly bright areas at individual locations.

All initial acquisitions were then brought into the compensation algorithm, values for differences between each image and the mean image were calculated, and the cycle was superimposed. When superimposed to the 245th image, ${∆}_{245}=0.119\%$, ${∆}_{246}=0.118\%$, and ${∆}_{247}=0.118\%$. This satisfied the threshold condition and terminated the superimposition. Therefore, $\;\mu_{i+2}$ was the compensation surface P(x,y). These results allowed us to determine that the surface had eliminated the difference value of the luminance. The response error surface of the dark-and-weak-target simulator is shown in Figure 10.

In Figure 10, it can be seen that the obtained response error surface is not flat and exhibits a trend similar to a Gaussian distribution; this may be explained by the fact that vignetting of an optical system itself characterized by Gaussian-like distribution, while the phenomenon of light leakage in LCD display panels is characterized by a random distribution.

### 5.3. Uniformity Compensation

The response error surface was used to compensate the output image of the dark-and-weak-target simulator to obtain $G_{i}^{\prime }(x,y)$, and a secondary acquisition was performed to obtain a grayscale SDF comparison curve for the acquired image before and after compensation, as shown in Figure 11. It can be seen in the figure that there is a significant difference in the grayscale SDF values of the acquired images before and after compensation, with SDF values of the acquired images before compensation continuing to rise with increases in grayscale values, finally reaching a high level. After compensation, the overall reduction in the grayscale SDF values is about 50%, and the curve tends to be flat. Figure 12 shows a histogram of gray levels of a captured image with a grayscale value of 200 before and after compensation. It can be seen that, overall, the pixels of the captured image converge to an ideal linear response relationship after compensation, and any differences between the pixel points with respect to luminescence response are largely eliminated.

### 5.4. Discussion

It can be seen that the grayscale SDF curve of the captured image after compensation is significantly lower than the grayscale SDF curve of the initially captured image. The difference is not significant at grayscale values below 109 because less light is transmitted from the LCD display panel in this grayscale range, and image-plane irradiance does not change significantly. However, when the grayscale value reaches 109, the influence of various factors on the energy uniformity of the image surface gradually becomes larger, so that the compensation effect becomes obvious, and the two curves begin to exhibit significant differences. The grayscale SDF curve of the initially captured image shows a steady increase, reaching a maximum value of 3.3% at a grayscale value of 255, while the grayscale SDF curve of the compensated captured image reaches a peak value of 1.42% at a grayscale value of 120 and then starts to decrease, falling to 0.49% at a grayscale value of 255. Additionally, Figure 12 clearly shows that the compensation algorithm was able to accurately concentrate and correct an original grayscale distribution which was severely deviated and scattered, and thus greatly improved the uniformity of the image surface energy.

Overall, by using an adaptive compensation algorithm to compensate for the output image of the dark-and-weak-target simulator, the grayscale SDF value of the sensor acquisition image was reduced by about 50% overall, and the image energy distribution converged with the ideal linear response relationship. At the same time, compared with the traditional method, this method is easier to calculate, the compensation effect is obvious, and does not need to rely on additional information for assistance.

## 6. Conclusions and Outlook

In the study reported in this paper, we used an adaptive compensation algorithm based on pixel-by-pixel fitting to effectively deal with the full-field-of-view energy inhomogeneity problem which affects dark-and-weak-target simulators as a result of light-source nonuniformity, the leakage of light from LCD display panels, and the vignetting of collimating optical systems. The experimental results proved that, after compensation, the grayscale SDF curve of the image was reduced by about 50% in general, which effectively eliminated the effects of uneven light sources, light leakage from the LCD display panel, and gradual vignetting of the optical system. Using our method, the uniformity of the energy of the image surface was greatly improved, and this may be seen as laying a foundation for high-quality image processing and feature extraction.

Researchers may now seek to optimize light distribution at the hardware level by dividing light sources into zones and regulating light for different regions. Image uniformity might be further improved by studying stationary-point image correction algorithms, as well as by image processing at the software level. Indeed, comprehensive zonal regulation of the light source and stationary-point image correction are two methods which are expected to deliver further control and correction of full-field-of-view energy distribution, and thus address still more comprehensively the problem of full-field-of-view energy nonuniformity which affects dark-and-weak-target simulators.

## Figures and Tables

**Figure 1 sensors-24-04147-f001:**
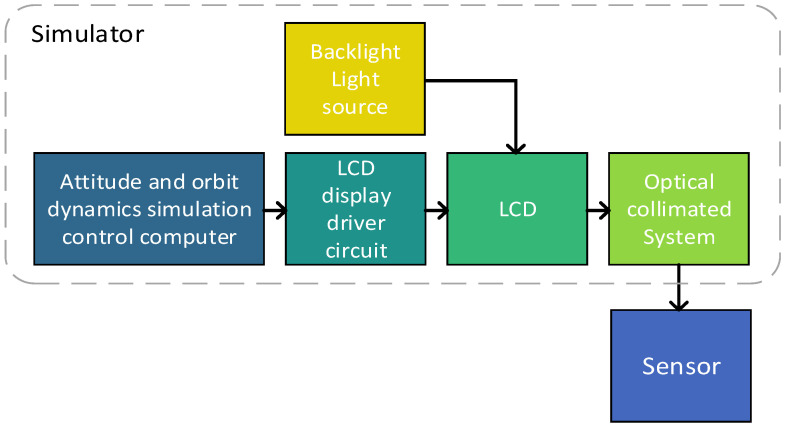
Composition of the dark-and-weak-target simulator.

**Figure 2 sensors-24-04147-f002:**
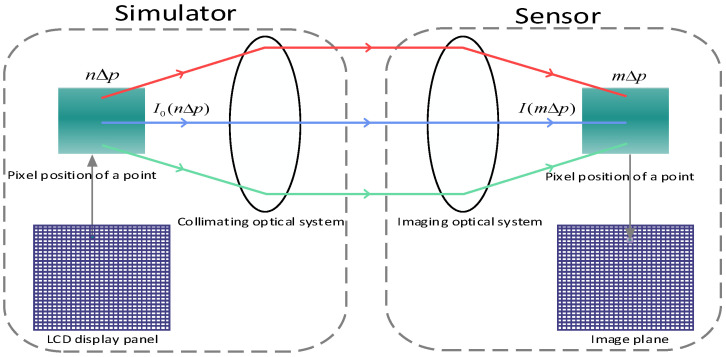
A model for energy transfer in dark-and-weak-target simulators.

**Figure 3 sensors-24-04147-f003:**
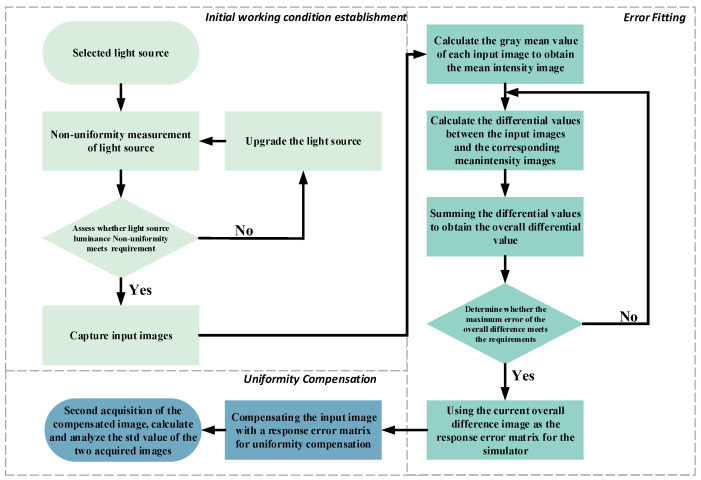
Flowchart of energy compensation process for dark/weak targets.

**Figure 4 sensors-24-04147-f004:**
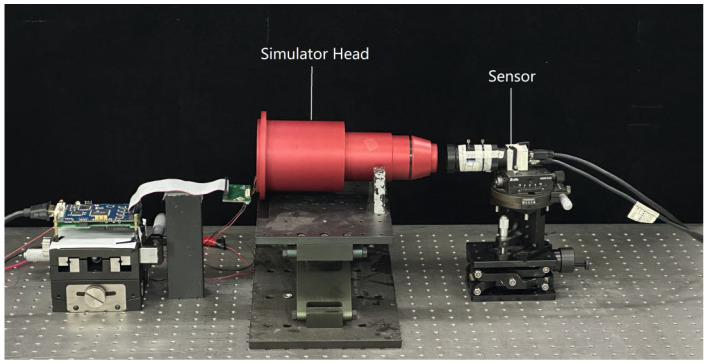
Test environment.

**Figure 5 sensors-24-04147-f005:**
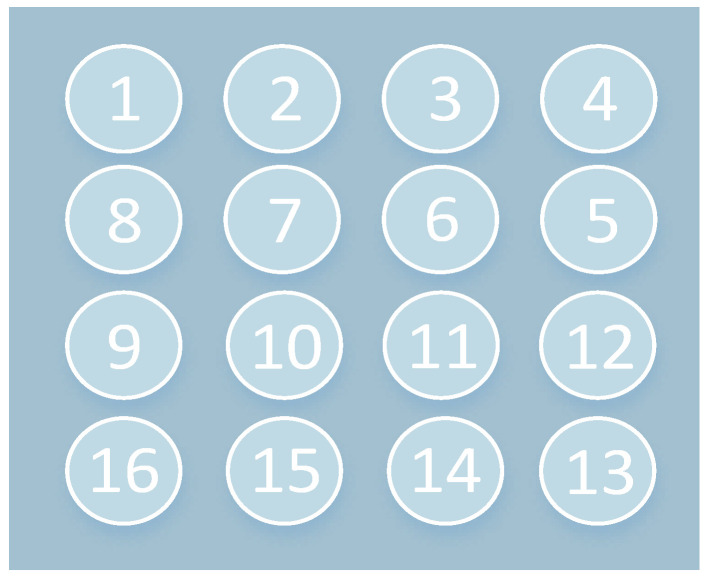
Numbering of sample positions on irradiated surfaces.

**Figure 6 sensors-24-04147-f006:**
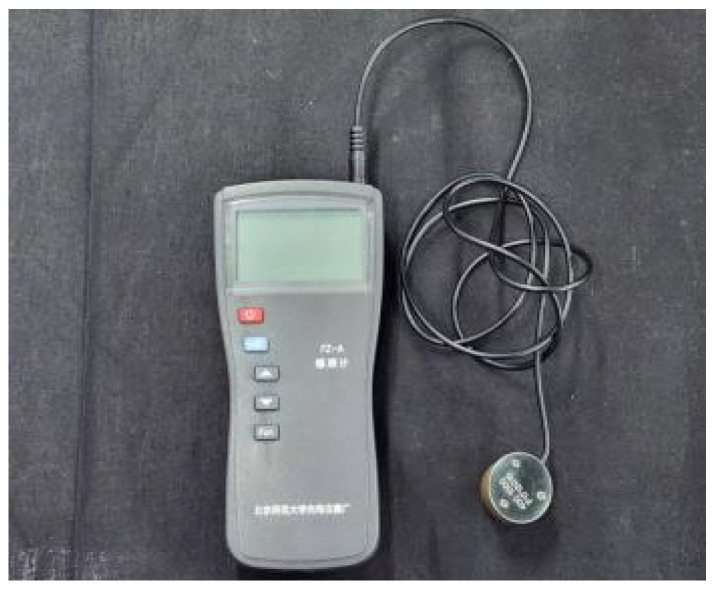
Irradiance meter.

**Figure 7 sensors-24-04147-f007:**
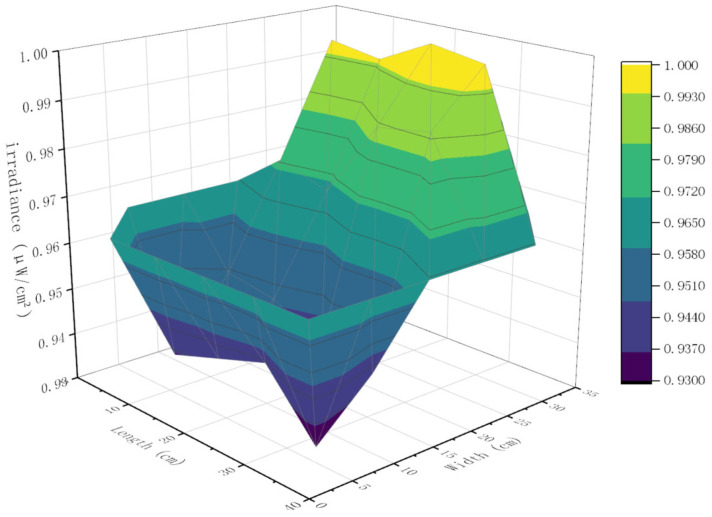
Backlight source irradiance distribution surface.

**Figure 8 sensors-24-04147-f008:**
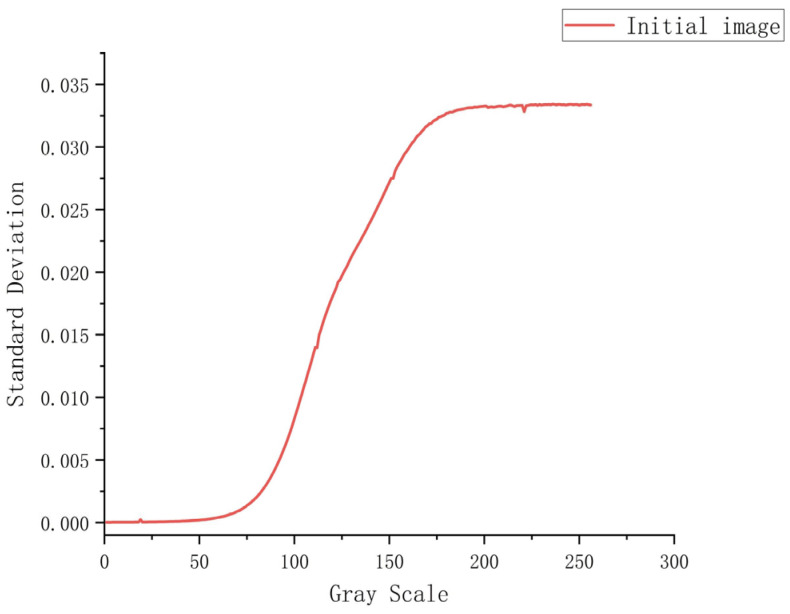
Grayscale SDF curves for initial-acquisition images.

**Figure 9 sensors-24-04147-f009:**
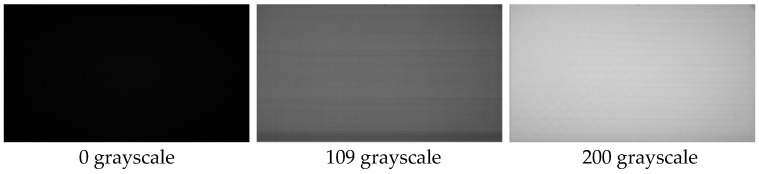
Initial-acquisition maps with different shades of gray.

**Figure 10 sensors-24-04147-f010:**
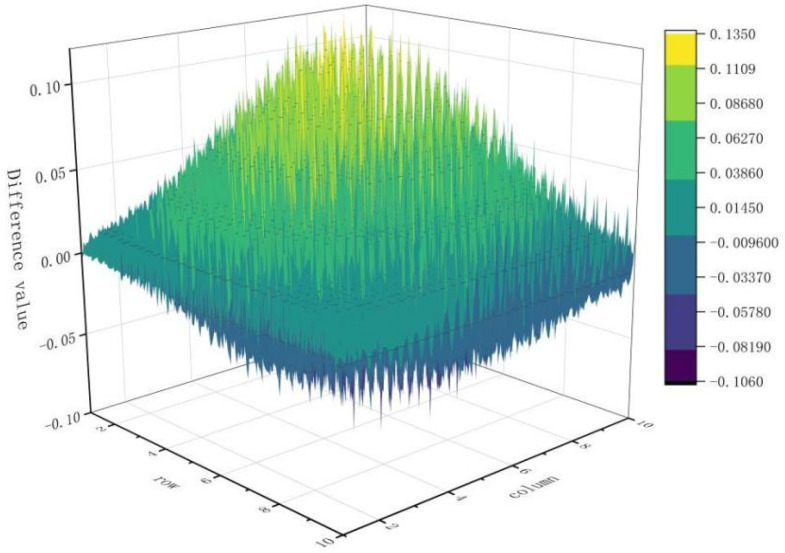
Response error surface.

**Figure 11 sensors-24-04147-f011:**
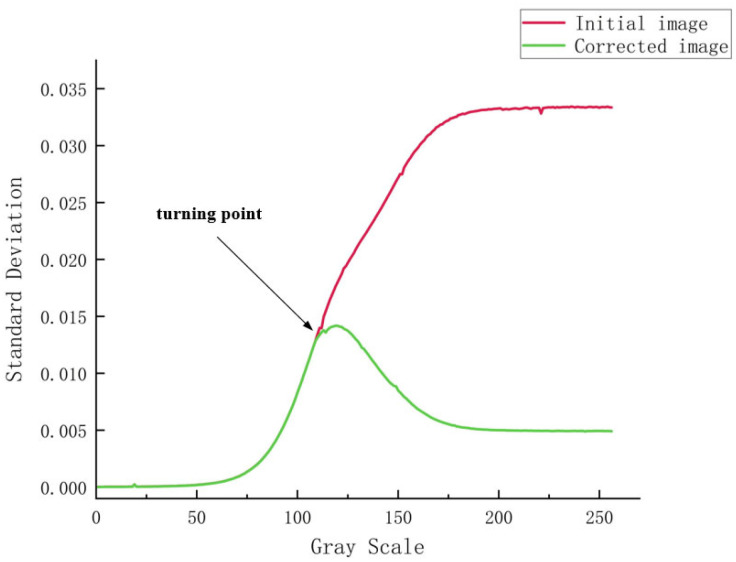
Grayscale SDF curves for initial image and compensated image.

**Figure 12 sensors-24-04147-f012:**
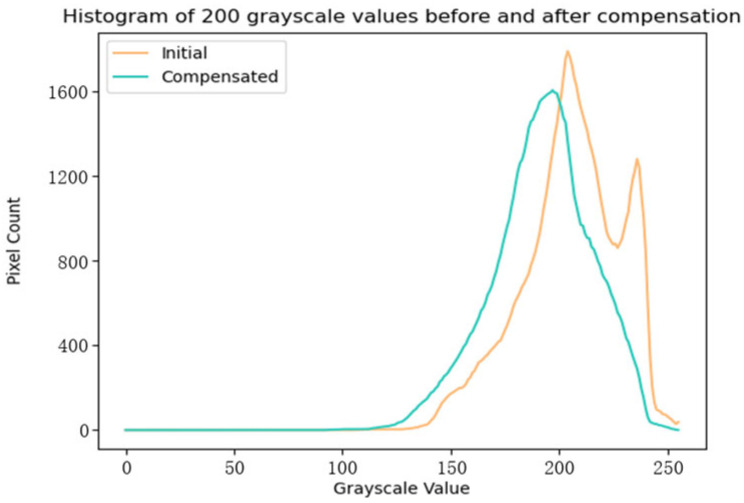
Histogram of 200-grayscale image before and after compensation.

**Table 1 sensors-24-04147-t001:** Irradiation inhomogeneity test results.

Test Point	Irradiance (µW/cm^2^)	Test Point	Irradiance (µW/cm^2^)
1	50.9	9	52.9
2	51.1	10	52.7
3	51.3	11	53.6
4	50.6	12	53.7
5	51.2	13	54.3
6	51.3	14	54.4
7	51.6	15	54.1
8	51.7	16	54.2

## Data Availability

The data in this study have been approved for publication.

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
