# Peer review of "A Compensation Method for Full-Field-of-View Energy Nonuniformity in Dark-and-Weak-Target Simulators"

_sensors, 2024, doi:10.3390/s24134147_

Round 1

Reviewer 1 Report

Comments and Suggestions for Authors

Dark-and-weak-target simulators are used as ground-based calibration devices to test and calibrate the performance metrics of star sensors. This paper established an energy transfer model of a dark-and-weak-target simulator based on the propagation of a point light source, and proposed a self-adaptive compensation algorithm based on pixel-by-pixel fitting.

Here are my suggestions.

(1)    For the experimental validation according to what are the initial working conditions set? Has it been set considering the actual situation of the application scenario? And can you add more details about the application conditions of the proposed method?

(2)    Can you make some comparisons between the proposed method and other traditional methods?

(3)    In figure 11, colors of the initial and the corrected image can be changed to make their difference is more significant.

Reviewer 2 Report

Comments and Suggestions for Authors

The manuscript presents a novel adaptive compensation algorithm designed to address nonuniform energy distribution in dark-and-weak-target simulators for star sensors. The work is technically sound and demonstrates significant innovation, particularly in the formulation and application of the energy transfer model. The methodology is well-detailed, with comprehensive mathematical formulations and experimental validation. Overall, the study makes a valuable contribution to improving the accuracy and reliability of star sensors, with practical implications for spacecraft navigation. However, the presentation quality of the manuscript needs improvement for publication.

1. Could the authors further clarify the boundary treatment applied in the model?

2. The authors have included many parameters in the theoretical part. It would be helpful to enhance the corresponding indications in the experiment section to facilitate readers’ understanding.

3. Regarding the presentation of the manuscript:

---The length-to-width ratio of Figure 2 is incongruous.

--- It would be better not to place Figure 5 and Figure 6 in the same row.

---Figure 7 is blurry.

---Please notice the font type of the equation numbers.

Comments on the Quality of English Language

Moderate editing of English language required

Round 2

Reviewer 2 Report

Comments and Suggestions for Authors

This manuscript can be accepted for publication.